# Macromolecular crowding and super-saturation protect hemodialysis patients from the onset of dialysis-related amyloidosis

Kichitaro Nakajima[1,8,9], Keiichi Yamaguchi[1,8,9], Masahiro Noji[2,9], César Aguirre[3], Kensuke Ikenaka [3], Hideki Mochizuki [3], Lianjie Zhou [4], Hirotsugu Ogi[4], Toru Ito[5], Ichiei Narita[5], Fumitake Gejyo[6], Hironobu Naiki[7], Suguru Yamamoto [5] ✉ & Yuji Goto [1,8] ✉

Dialysis-related amyloidosis (DRA), a serious complication among long-term hemodialysis patients, is caused by amyloid fibrils of β2-microglobulin (β2m). Although high serum β2m levels and a long dialysis vintage are the primary and secondary risk factors for the onset of DRA, respectively, patients with these do not always develop DRA, indicating that there are additional risk factors. To clarify these unknown factors, we investigate the effects of human sera on β2m amyloid fibril formation, revealing that sera markedly inhibit amyloid fibril formation. Results from over 100 sera indicate that, although the inhibitory effects of sera deteriorate in long-term dialysis patients, they are ameliorated by maintenance dialysis treatments in the short term. Serum albumin prevents amyloid fibril formation based on macromolecular crowding effects, and decreased serum albumin concentration in dialysis patients is a tertiary risk factor for the onset of DRA. We construct a theoretical model assuming cumulative effects of the three risk factors, suggesting the importance of monitoring temporary and accumulated risks to prevent the development of amyloidosis, which occurs based on supersaturation-limited amyloid fibril formation in a crowded milieu.

Dialysis-related amyloidosis (DRA), one of the systemic amyloidoses[1,2], is a serious complication of long-term hemodialysis therapy and a threat to the health of dialysis patients[3–8]. The pathological hallmark of DRA is the deposition of amyloid fibrils of β2-microglobulin (β2m) in the peritenons and synovial membranes of the carpal tunnel. β2m is a component of the major histocompatibility complex (MHC) class I expressed on the surface of all nucleated cells[9]. β2m monomers are released into the bloodstream when MHCs dissociate. In individuals with healthy kidneys, most of the released β2m monomers are eliminated from the bloodstream by glomerular filtration and subsequent reabsorption and catabolism by the proximal tubules, keeping the serum β2m concentration at ~1 μg/mL. However, in long-term hemodialysis patients, the concentration of β2m increases up to ~50 μg/mL because it is not degraded due to kidney failure and is not completely eliminated by maintenance dialysis treatments. Consequently, β2m forms amyloid fibrils and causes DRA in patients who have been on hemodialysis therapy for more than 10 years. To date, only two cases of naturally occurring β2m mutants have been reported: D76N in non-dialysis patients[10] and V27M in a DRA patient[11]. These facts indicate that a high serum β2m concentration and long

A list of author affiliations appears at the end of the paper. ✉e-mail: yamamots@med.niigata.ac.jp; gtyj8126@protein.osaka-u.ac.jp

dialysis vintage are the primary and secondary risk factors, respectively, for the onset of DRA. However, patients with these risks do not always develop DRA[4–8], indicating that there are additional risk factors (Supplementary Fig. 1).

In Japan, dialysis technology has progressed markedly in the past 40 years and has extended the time for first-time carpal tunnel surgery (CTS), as a proxy for the onset of DRA[6,8]. A comparison of cohorts of chronic hemodialysis patients (more than 200,000 in total) in 1998 and 2010 showed that the risk for DRA decreased from 1998 to 2010[6]. The risk of first-time CTS was almost halved, even though it was still 10% in the 2010 cohort with a dialysis vintage longer than 20 years. The survey indicated that improvement in dialysis technology, particularly the introduction of biocompatible dialysis membranes, contributed to the decreased risk[6]. Recently, a retrospective survey of 222 patients across 4 periods from 1982 to 2019 showed that improvement of β2m clearance via advances in dialysis technology contributed to extending the time between starting hemodialysis and the first surgery for CTS[8]. The time was 12.4 years in period 1 (1982–1989) but extended to 22.4 years in period 4 (2010–2019).

Among various amyloidogenic proteins, β2m, a β-barrel globular protein consisting of 99 amino acid residues, is particularly useful for addressing the relationship between protein folding and misfolding (i.e., amyloid fibril formation)[10,12–18]. We have studied β2m amyloid fibril formation[19–24] from the physicochemical viewpoint, concluding that amyloid fibril formation is similar to crystallization of solutes[22,25–27]. According to this view, amyloid fibrils are crystal-like aggregates of denatured proteins, which are formed above solubility upon breaking supersaturation. Supersaturation, a fundamental phenomenon of nature with underlying physicochemical mechanisms remaining elusive[28–30], is required for crystallization and is involved in numerous phenomena, e.g., the supercooling of water prior to ice formation or various types of lithiasis caused by small organic compounds[22]. The same will be true for crystal-like amyloid fibrils. Under supersaturated conditions, an unknown trigger can break supersaturation, leading to amyloid fibril formation[20,22,31].

To gain insights into the pathogenesis of DRA under macromolecular crowding, we studied the effects of serum components on β2m amyloid fibril formation. It should be noted that, although previous studies[32–35] investigated amyloid fibril formation in a crowded milieux (e.g., serum milieu), there is no consensus on the effects of macromolecular crowding in a serum milieux. With HANABI-2000[36–38], an ultrasonication-forced amyloid fibril inducer, the effects of sera from hemodialysis patients on β2m amyloid fibril formation were investigated at 60 °C where the reactions were accelerated. To perform a systematic investigation, we collected over 100 sera from individuals with or without hemodialysis therapy. The results showed that sera inhibited β2m amyloid fibril formation based on the macromolecular crowding effects arising from serum albumin, the most abundant protein in sera, and that the inhibitory effects were weaker for hemodialysis patients, particularly before maintenance dialysis treatments, because of the decreased serum albumin concentration. We modelled the development of DRA assuming that the primary, secondary, and tertiary risk factors are an increased β2m concentration, a long dialysis vintage, and decreased serum albumin concentration, respectively. In the model, the effects of temperature were consolidated, allowing us to discuss the onset of DRA by extrapolating experimental results acquired under the non-physiological temperature (60 °C) to the physiological one (37 °C). The unified model suggested that accumulated risks break the extracellular proteostasis network[39] under supersaturation, leading to amyloid fibril formation and thus the development of amyloidosis. The current model based on solubility- and supersaturation-limited amyloid fibril formation will be applicable to a variety of amyloidoses and, furthermore, will be useful for devising therapeutic strategies.

## Results

### HANABI assay for β2m amyloid fibril formation

Previously, Noji et al.[23,24] reported that β2m monomers even at a neutral pH form amyloid fibrils at high temperature under agitation, effective conditions to break supersaturation. Thus, control reactions were performed at 1.0 mg/mL β2m at pH 7.4 and 60 °C. To apply further effective agitation to induce amyloid fibril formation, we adopted an originally developed ultrasonic instrument, HANABI-2000[36–38], which accelerates amyloid fibril formation[19,20,40,41] by the effects of ultrasonic cavitation[42,43] and monitors amyloid formation by amyloid-specific fluorescence dye, thioflavin-T (ThT)[44]. Scheme of the experiment is described in Supplementary Fig. 2. Consistent with previous reports[23,24], β2m solutions did not show an increase in ThT fluorescence intensity without ultrasonication, but did when ultrasound was applied (Fig. 1a), indicating amyloid formation under ultrasonication.

After the ultrasonic experiments, the aggregates formed were analyzed. The circular dichroism (CD) spectra showed a transition from native monomers with a minimum at 220 nm to amyloid fibrils with a minimum at 230 nm (Fig. 1b), consistent with previous reports[21,23,24]. It should be noted that, some of the β2m monomers were thermally denatured at 60 °C, but the denatured monomers were not dominant as at 95 °C (Fig. 1b). The fraction of denatured monomers at 60 °C was -14.3%, which was estimated from the thermal denaturation curve (Fig. 1c). High performance reversed-phase chromatography revealed the persistence of intact monomers (Fig. 1d, Supplementary Note 1, and Supplementary Fig. 3), confirming that no fragmentation or other damage of β2m monomers occurred during the ultrasonic experiments. Transmission electron microscopy (TEM) images of aggregates also indicated a fibrous morphology (Fig. 1e, Supplementary Note 2, and Supplementary Fig. 4a). However, since TEM images of amyloid fibrils produced under ultrasonication often showed poor contrast because of the extensive fragmentation[20,45] and unknown effects, we performed a seeding reaction to prepare longer amyloid fibrils. The seeded amyloid fibrils clearly showed a fibrous morphology with improved contrast (Fig. 1f, Supplementary Note 2, and Supplementary Fig. 4b).

To investigate the effects of human sera on amyloid fibril formation, we then added an aliquot of sera collected from a non-dialysis control without dialysis therapy (DT(−), #9) to the standard β2m solution (Fig. 1a). At a serum concentration of 2.5% (v/v), although ThT fluorescence increased, the lag time was much longer, and the final ThT fluorescence intensity was low. At a serum concentration of 15% (v/v), ThT fluorescence remained unchanged within 15 h. TEM images without a clear fibrous morphology suggested the formation of amorphous aggregates (Fig. 1g, Supplementary Note 2, and Supplementary Fig. 4c). These results indicate that the addition of sera inhibited β2m amyloid fibril formation.

### Concentration-dependent inhibitory effects of sera

To systematically investigate the inhibitory effects of sera on β2m amyloid fibril formation, three types of sera were prepared: control sera from individuals without dialysis therapy (DT(−), #9), and sera collected from dialysis patients immediately before (DT(+, Pre), #8) and after (DT(+, Post), #8) maintenance dialysis treatments. All dialysis patients examined in this study underwent maintenance dialysis treatments involving 4–5 h sessions conducted three times a week (see Methods). The effects of sera on amyloid formation were assayed at varying serum concentrations, 0.3–15% (v/v) (Fig. 2a–c). Here, β2m monomers from sera were less than 1% of the recombinant β2m monomers in the standard solution (1.0 mg/mL), indicating that the effects of β2m monomers present in sera were negligible (Supplementary Note 3 and Supplementary Fig. 5). In other words, we eliminated the primary risk factor to focus on other risk factors. The kinetics monitored by ThT fluorescence were analyzed using two parameters, a lag time and the maximum intensity of ThT fluorescence, where the lag

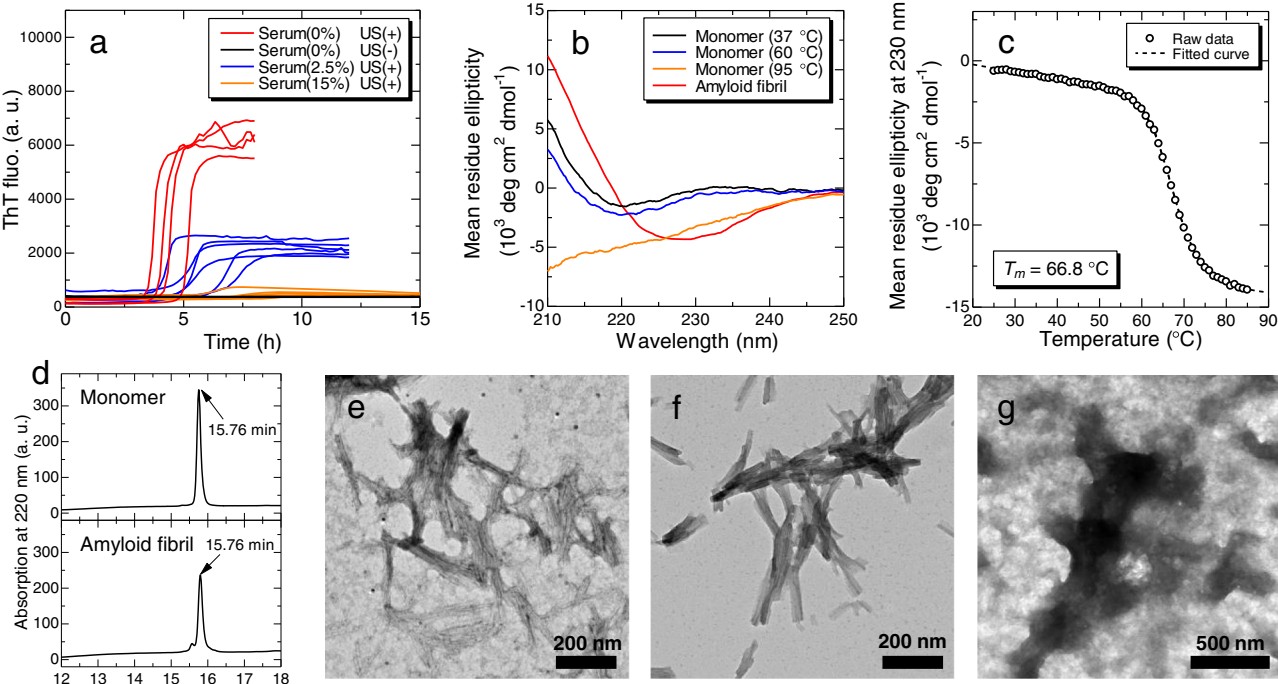

**Fig. 1 | HANABI assay for β2m amyloid fibril formation. a** ThT fluorescence kinetics of β2m solutions at various conditions ($n \geq 4$). Serum was collected from a non-dialysis control #9 (DT(−)). **b** CD spectra of the initial monomers at 37, 60, and 95 °C, and amyloid fibrils formed by ultrasonication. **c** Thermal denaturation curve of β2m monomer monitored by the CD ellipticity at 230 nm. $T_m$ is the denaturation midpoint obtained from the fitted curve. **d** High performance reversed-phase chromatograms for intact monomers (upper) and amyloid fibrils formed by ultrasonication (lower). TEM images of (**e**) ThT-positive aggregates formed by ultrasonication, (**f**) aggregates seeded and elongated from the amyloid fibrils formed by ultrasonication, and (**g**) ThT-negative aggregates formed in the presence of 15% (v/v) serum. In TEM observations, five images were acquired for each sample. The images in panel (**e**, **f**, and **g**) are the representative image, and remaining four images are shown in Supplementary Fig. 4. a. u., arbitrary units. Source data are provided as a Source Data file.

time was when the ThT fluorescence intensity became one tenth of the maximum.

For all types of sera, the higher the concentration of sera added, the longer the lag time (Fig. 2d) and lower the ThT fluorescence intensity (Fig. 2e). It was initially unclear whether the lower ThT intensity represents a decreased amount of amyloid fibrils or an apparent decrease due to optical absorption of incident beams by serum components, etc. To investigate the relationship between ThT fluorescence intensity and the amount of amyloid fibrils, a series of seeding experiments were conducted using products obtained at various serum concentrations (Supplementary Note 4 and Supplementary Fig. 6). The lag time of the ThT kinetics verified by the seeding experiments showed an inverse correlation with the initial ThT fluorescence intensities of the seed solution. Moreover, when serum was added to the preformed amyloid fibrils, the decrease in the ThT fluorescence intensity due to optical absorption by serum components was much less than the decrease caused by amyloid fibril formation in the presence of serum (Fig. 2f). These results indicate that, even in the presence of serum, the maximal ThT fluorescence intensity was proportional to the amount of amyloid fibrils, and that the decrease in ThT fluorescence maximum intensity represented the inhibitory effects of serum components.

## Deterioration of inhibitory effects by long-term dialysis

To focus on the effects of dialysis vintage, we examined the difference in the effects of sera with and without hemodialysis therapy for a total of 60 sera at different dialysis vintages (Supplementary Fig. 7). As shown in Fig. 3a, 30 of the 60 sera were collected from dialysis patients (DT(+, Pre), $N = 30$), and the remaining 30 sera were collected from non-dialysis controls (DT(−), $N = 30$). The sera from dialysis patients were collected before maintenance dialysis treatments, and the serum

concentration added was 5% (v/v), where the serum-dependent differences in the lag time and ThT fluorescence intensity were large (Fig. 2d, e). The representative kinetics are shown in Fig. 3b. Without serum addition, the ThT kinetics indicated rapid amyloid fibril formation with a lag time of approximately 2 h and a marked increase in the ThT fluorescence intensity, confirming the rapid formation of a large amount of amyloid fibrils. Although the two types of sera markedly inhibited amyloid formation regarding both the lag time and ThT fluorescence intensity, the degree of inhibition was different between the two groups. The average lag times were 3.9 and 5.4 h for DT(+, Pre) and DT(−) groups, respectively (Fig. 3c), and the maximal ThT fluorescence intensities were 6600 and 2300 for DT(+, Pre) and DT(−) groups, respectively (Fig. 3d). These results suggest that, even after eliminating the primary risk of an increased β2m concentration, sera from dialysis patients retained an additional risk of amyloid fibril formation in terms of reducing serum-dependent inhibitory effects.

A previous study[46] showed that the secondary risk factor for the onset of DRA, following the primary risk factor of an increased serum β2m level, is a long dialysis vintage[4–6,8]. The relationship between the dialysis vintage and lag time (Fig. 3e) or ThT fluorescence intensity (Fig. 3f) revealed a minimal correlation between them. The results suggest that, although the temporal risks, other than the increased β2m concentration, notably increased in dialysis patients, they did not change markedly during the dialysis vintage. Given that the dialysis vintage is a strong risk factor for the onset of DRA, accumulated (or integrated) temporal risks over the dialysis vintage may determine the onset of DRA. In other words, the results of HANABI assays provide an indicator of the temporal risks of DRA onset. It is likely that a combination of temporal risks, an increased β2m concentration, and dialysis vintage collectively provides a susceptibility risk biomarker[47] for DRA, which represents the potential for the

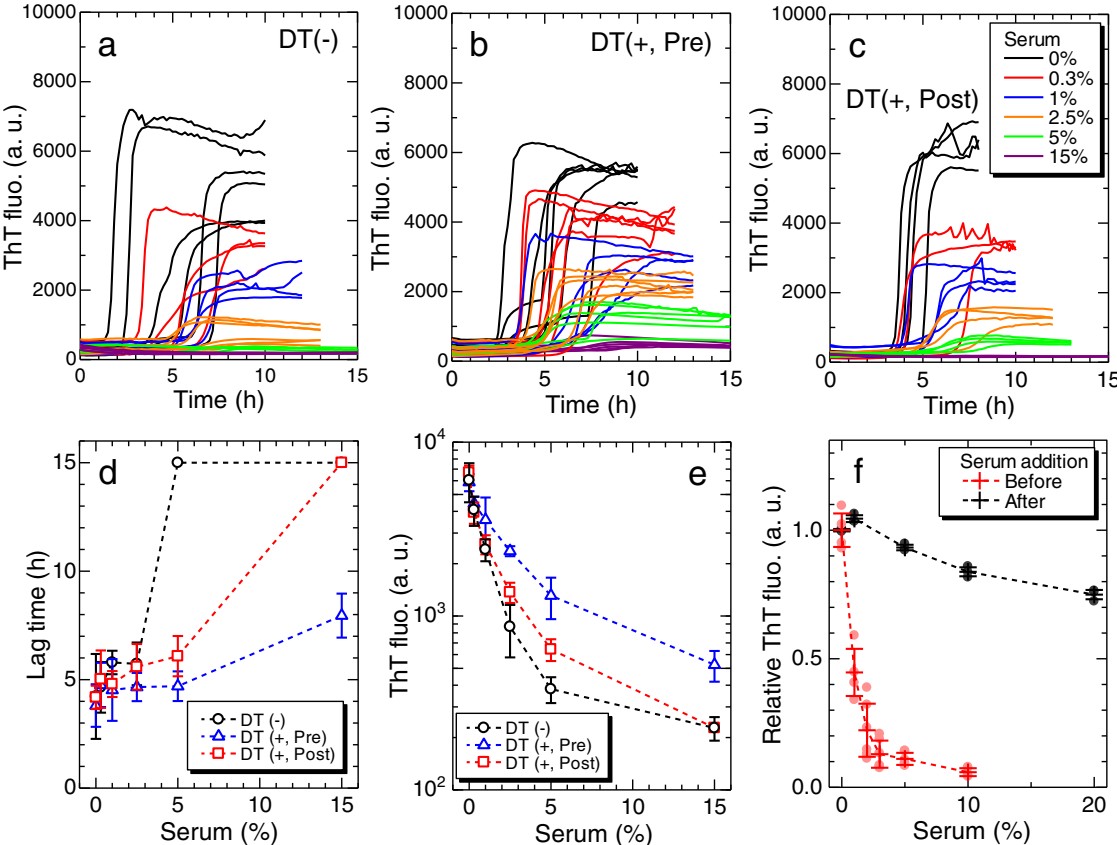

**Fig. 2 | Concentration-dependent inhibitory effects of sera on β2m amyloid fibril formation.** ThT fluorescence kinetics at varying concentrations of sera collected from (**a**) a non-dialysis control (DT(−), #9) and a dialysis patient immediately (**b**) before (DT(+, Pre), #8) and (**c**) after maintenance dialysis treatment (DT(+, Post), #8), respectively. For each condition, measurement was performed multiple times for independent solutions ($n \geq 4$). Dependence of (**d**) the lag time and (**e**) ThT fluorescence intensity on serum concentration for each serum ($n \geq 4$). **f** Decrease in the ThT fluorescence intensity with the addition of serum to the sample solution before and after amyloid fibril formation ($n = 3$). In panels (**d**, **e**, and **f**), the centers for the error bars and error bars denote the mean and standard deviation among independent sample solutions, respectively. a. u., arbitrary units. Source data are provided as a Source Data file.

onset of disease in an individual who does not currently have clinically apparent symptoms.

## Amelioration of inhibitory effects by maintenance dialysis

Next, we examined the effects of maintenance dialysis treatments, conducted three times a week, on the inhibitory effects of serum. As shown in Fig. 4a, maintenance dialysis treatment markedly changes the serum milieu: the β2m monomers are eliminated approximately more than 50%, and body weights of patients are decreased because of the expulsion of water. Sera were collected from a total of 28 dialysis patients immediately before (DT(+, Pre)) and after (DT(+, Post)) maintenance dialysis treatments, and HANABI assays were conducted. The patients were different from those used to examine the effects of dialysis vintage. Notably, the maintenance dialysis treatments markedly recovered the inhibitory effects of serum against amyloid formation (Fig. 4b). Statistical analysis (Supplementary Note 5 and Supplementary Fig. 8) confirmed the ameliorating effects in terms of a longer lag time (Fig. 4c) and lower ThT fluorescence intensity (Fig. 4d). These results indicate that, although the inhibitory effects of sera were deteriorated in long-term dialysis patients, they were ameliorated by maintenance dialysis treatments in a short term.

Importantly, a single maintenance dialysis treatment eliminates a large amount of extra water (usually ~5% of body weight) from a patient's body, largely from blood, along with biological waste, playing the role of the kidney in patients with end-stage renal failure. The weight change of patients studied in the present study ranged from 2 to 8% (Fig. 4e, f). Although plasma refilling occurs to maintain the blood volume during dialysis treatments[48], the concentrations of serum components transiently and immediately increase after maintenance dialysis due to the expulsion of water. This is likely to recover the inhibitory effect, resulting in a decrease in the temporary risk (*TR*) (see Discussion).

## Serum albumin is primarily responsible for the inhibitory effects

To reveal the underlying mechanism of the change in the inhibitory effects of sera, correlations between concentrations of 27 serum components and amyloidogenicities in terms of the lag time or ThT fluorescence intensity were systematically investigated (Supplementary Fig. 9). We included data for both calculation of the Pearson correlation coefficient between the concentration of each serum component and the lag time or ThT fluorescence intensity. The correlations suggested three candidates of serum components that markedly affect amyloid fibril formation: serum albumin, ureic acid, and creatinine (Supplementary Note 6). We separately investigated the effects of the three candidates, revealing that only serum albumin significantly inhibited amyloid formation (Supplementary Fig. 10). Lower albumin concentrations tended to induce faster kinetics and larger amounts of amyloid fibrils (Supplementary Fig. 9, panel 23), indicating that serum albumin plays a major role in inhibiting β2m amyloid formation in a serum milieu.

To address amyloid fibril formation in a crowded milieu, we performed HANABI assays in the presence of 5 mg/mL serum albumin or 5 mg/mL polyethylene glycol (PEG) (Fig. 5a–c). Here, we selected PEG as a negative control in the HANABI assay and quartz crystal

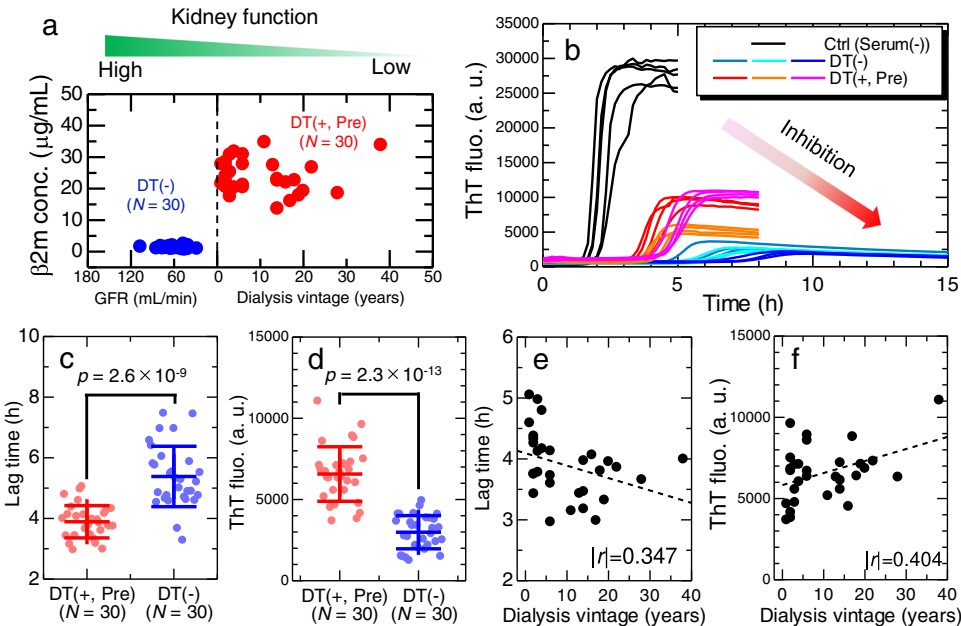

**Fig. 3 | Deterioration of inhibitory effects by long-term dialysis. a** Clinical characteristics of non-dialysis control (DT(−), $N = 30$) and dialysis patient (DT(+, Pre), $N = 30$) groups. For the non-dialysis group, their glomerular filtration rates (GFR) are used as an index of kidney function, because they have never received dialysis treatments. **b** Representative ThT fluorescence kinetics using sera collected from non-dialysis controls and dialysis patients. For each sample, measurement was performed multiple times for independent solutions ($n \geq 4$). Comparison of the effect of sera with or without dialysis treatment on β2m fibril formation in terms of the (**c**) lag time and (**d**) ThT fluorescence intensity. The $p$-values were caluculated by the unpaired one-sided $t$-test. For all 60 samples, row data are shown in Supplementary Fig. 7. Error bars in panels (**c** and **d**) denote the standard deviation among independent serum samples ($N = 30$). Dialysis-vintage dependence of (**e**) the lag time and (**f**) ThT fluorescence intensity ($N = 30$). The $r$-values indicate the Pearson correlation coefficient. a. u., arbitrary units. Source data are provided as a Source Data file.

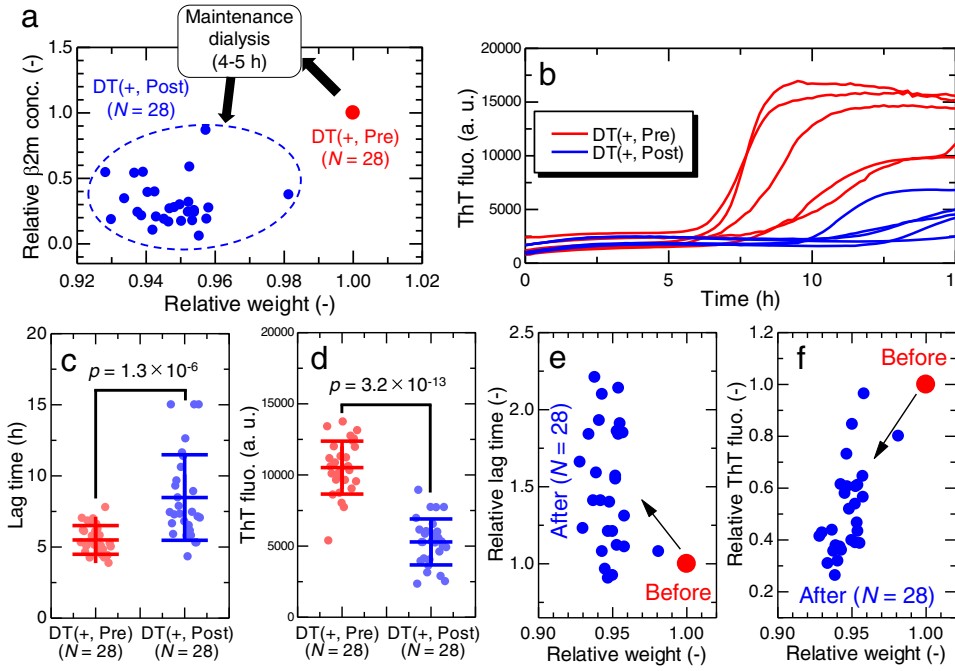

**Fig. 4 | Amelioration of inhibitory effects by maintenance dialysis. a** Effects of maintenance dialysis treatment on the serum β2m concentration and body weight of dialysis patients ($N = 28$), who donated their serum immediately before (DT(+, Pre)) and after (DT(+, Post)) maintenance dialysis treatment. The relative changes in the two parameters were shown with respect to their values before dialysis treatment. **b** ThT fluorescence kinetics of β2m fibril formation with 5% (v/v) addition of sera collected from an identical dialysis patient immediately before (DT(+, Pre), #15) and after (DT(+, Post), #15) a single maintenance dialysis treatment.

Significance examination for the effects of serum before and after dialysis treatment in terms of the (**c**) lag time and (**d**) ThT fluorescence intensity ($N = 28$). The $p$-values were calculated by the paired one-sided $t$-test. For all dialysis patients, row data are shown in Supplementary Fig. 8. Error bars in panels (**c** and **d**) denote the standard deviation among independent serum samples ($N = 28$). Correlation between change in relative weight and (**e**) lag time and (**f**) ThT fluorescence intensity. a. u., arbitrary units. Source data are provided as a Source Data file.

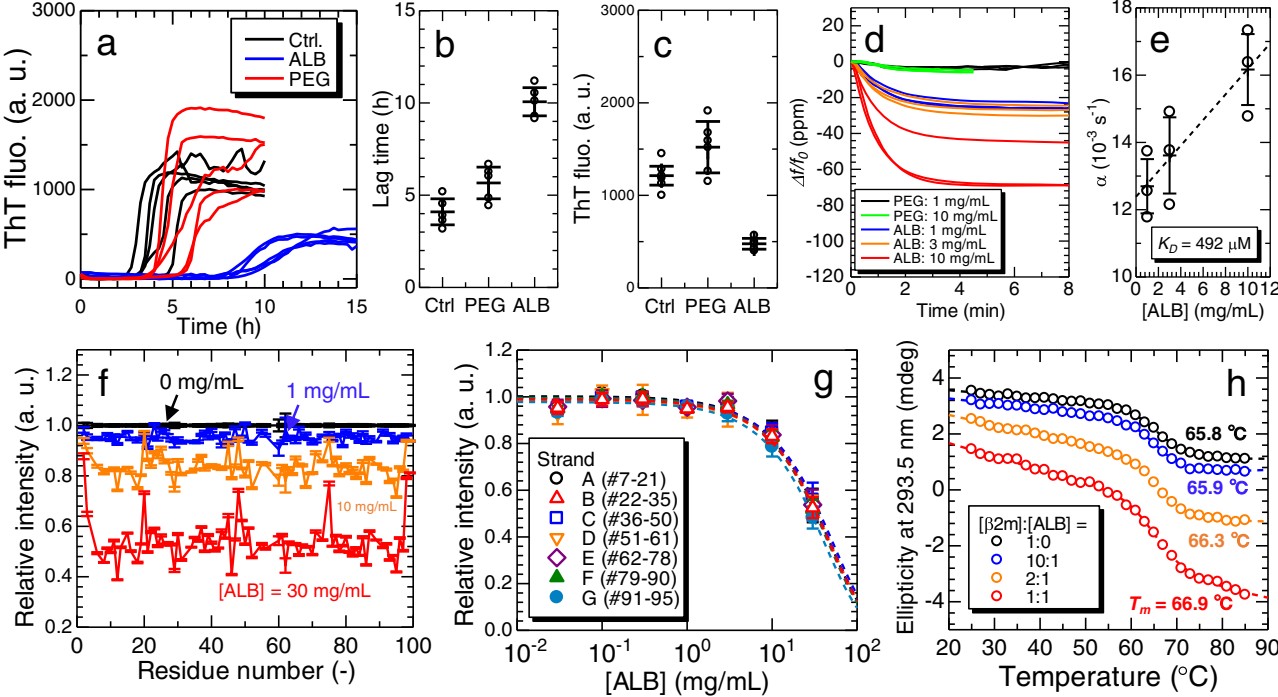

**Fig. 5 | Analysis on interaction between β2m monomer and serum albumin.**
**a**–**c** The effects of serum albumin (ALB) and polyethylene glycol (PEG) on β2m amyloid fibril formation. **a** ThT fluorescence kinetics in the presence of 5 mg/mL serum albumin (ALB) or polyethylene glycol (PEG) ($n = 5$), (**b**) their lag times, and (**c**) ThT fluorescence intensities. Centers for error bars and error bars denote the mean and standard deviation among independent sample solutions ($n = 5$), respectively. **d**, **e** Interactions between β2m monomer and serum albumin monitored by QCM experiments ($n = 3$): (**d**) Relative frequency change and (**e**) calculation of the dissociation constant, $K_D$, between β2m monomer and serum albumin. The error bars denote the standard deviation among independent measurements ($n = 3$). **f** Relative intensity changes in the NMR signal of the [15]N-labeled β2m monomer at different albumin concentrations. The error bars indicate errors in estimation of relative peak intensities. **g** Relative intensity changes in the NMR signals averaged over each strand. The error bars indicate errors in estimation of relative peak intensities. **h** Thermal denaturation curves of β2m monomers in the presence of serum albumin with various stoichiometries. $T_m$ value is a denaturation midpoint of β2m monomer at each serum albumin concentration. a. u., arbitrary units. Source data are provided as a Source Data file.

microbalance (QCM) measurement discussed below because of low affinity for proteins. The results showed that, although serum albumin markedly inhibited amyloid formation, PEG slightly extended the lag time, but the resulting amount of amyloid fibrils monitored by ThT fluorescence slightly increased. The mechanism of inhibition by serum albumin will be discussed in Discussion, along with results of interaction analysis between β2m monomer and serum albumin.

**Analysis on interaction between β2m monomer and serum albumin**
Interactions between β2m monomers and serum albumin were investigated by wireless-electrodeless QCM biosensor[49,50], solution nuclear magnetic resonance (NMR) measurements, and near-UV CD spectroscopy. In QCM analysis, serum albumin bound to β2m monomer in a concentration-dependent manner, but PEG did not because of low affinity for proteins (Fig. 5d). We considered that the simplest mechanism of complex formation existed between β2m monomers and serum albumin with a stoichiometry of 1:1. From the fitting analysis of frequency-change curves[51] (see Methods), the dissociation constant ($K_D$ value) between β2m monomer and serum albumin at 37 °C was calculated to be $K_D = 492$ μM (Fig. 5e). The $K_D$ values remained on the order of 100 μM even under high temperatures up to 60 °C (Supplementary Fig. 11), indicating that the interaction between β2m monomer and serum albumin did not change markedly from the physiological temperature (i.e., 37 °C) to that of HANABI assays (i.e., 60 °C).

NMR analysis showed the decrease in the NMR signal intensity of [15]N-labeled β2m monomers with an increasing serum albumin concentration (Fig. 5f). Since the β2m native structure consists of seven β-strands, the averaged relative intensity changes were calculated for respective β-strands and compared (Fig. 5g), indicating that there is no specific binding site in the β2m monomers. The dissociation constant estimated from the decrease in the NMR signal was ~500 μM, being on the same order as that obtained from QCM analysis. Although NMR measurements at high temperatures could not be performed owing to the limitations of experimental setup, it was inferred that the non-specific interactions persist at high temperatures because of subequal dissociation constants over a wide range of temperature as revealed by the QCM measurements. These results indicate that the interaction between β2m monomer and serum albumin is weak nonspecific binding[52].

To investigate the effects of serum albumin on the stability of β2m monomers, we observed thermal denaturation of β2m monomers in the presence of serum albumin by near-UV CD spectroscopy using a specific peak of native β2m at 293.5 nm (Supplementary Note 7 and Supplementary Fig. 12). The results revealed that the addition of serum albumin increased the $T_m$ value of the native β2m monomer ~1 °C (Fig. 5h), indicating that the thermostability of β2m monomer was nearly unchanged by the addition of serum albumin with the stoichiometry range of β2m:ALB = 10:1 to 1:1. Because the typical stoichiometry in the HANABI assays was β2m:ALB = 2.5:1, the inhibitory effect of serum albumin on β2m amyloid fibril formation was not directly explained by the stabilization of native β2m monomers. However, serum albumin is likely to stabilize the β2m monomers in serum where the stoichiometry is much higher (e.g., β2m:ALB = 1:120-1:4000), as discussed below.

## Discussion
The effects of molecular crowders on amyloid fibril formation are explained by three dominant mechanisms[32,33,53]: (i) volume exclusion

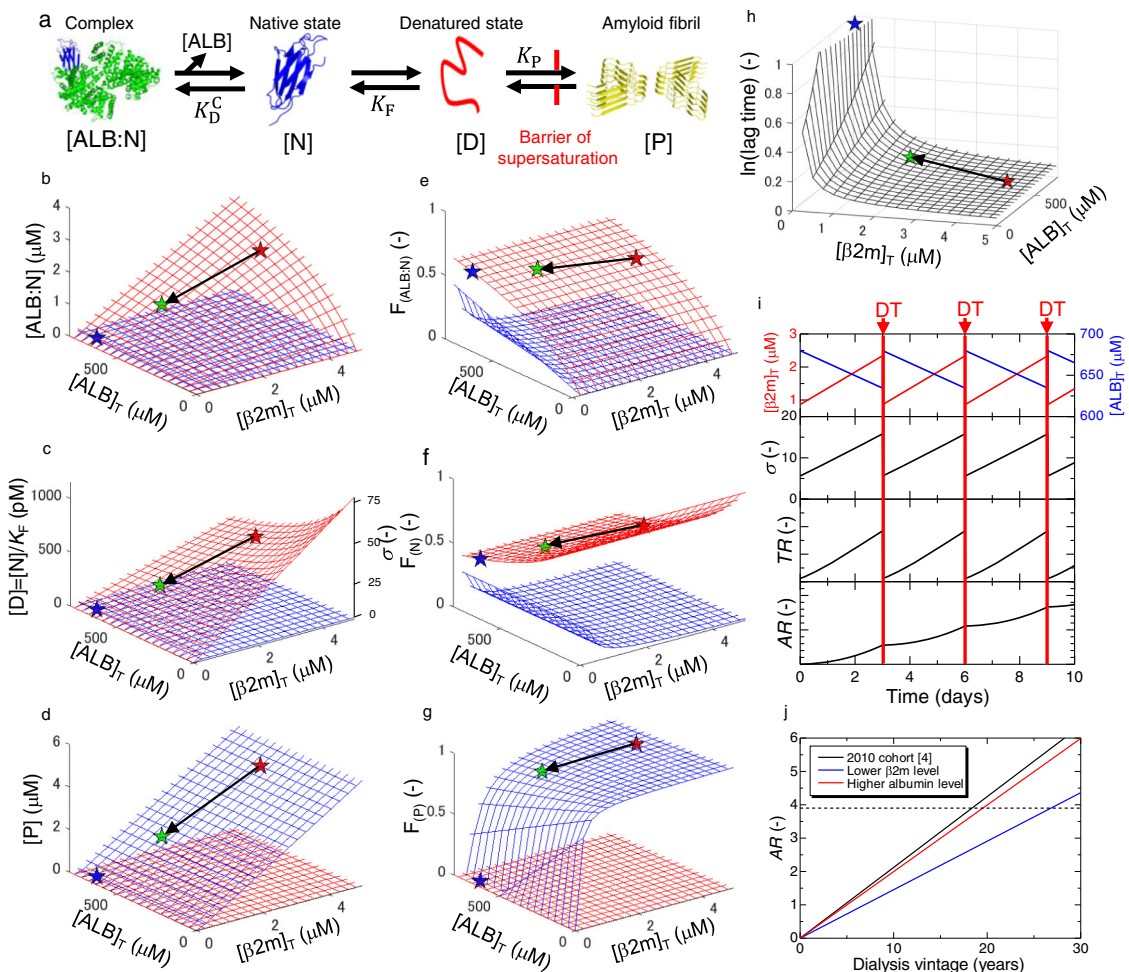

**Fig. 6 | The theoretical calculation of β2m amyloid fibril formation in a serum milieu. a** Schemes of β2m amyloid fibril formation in the presence of serum albumin. The molecular illustrations were depicted using PyMOL Molecular Graphics (ver. 2.5.1). **b–d** Dependences on total β2m ([β2m]$_T$) and serum albumin ([ALB]$_T$) concentrations of (**b**) native β2m-serum albumin complex, (**c**) denatured monomer, and (**d**) amyloid fibrils, before (red mesh) and after (blue mesh) breakdown of supersaturation. In panel (**c**), the axis of the supersaturation ratio, $\sigma$ = [D]$^S$/[D]$_C$, is also shown. **e–g** Dependences as shown in panels (**b–d**) are represented by fractions of (**e**) native β2m-serum albumin complex, (**f**) native monomer, and (**g**) amyloid fibrils. **h** The relative change in the lag time as a function of [β2m]$_T$ and

[ALB]$_T$. In panels (**b–h**), the blue, red, and green stars indicate the representative values for non-dialysis controls (DT(−), [β2m]$_T$ = 2 μg/mL (0.17 μM) and [ALB]$_T$ = 45 mg/mL (680 μM)) and dialysis patients before maintenance dialysis (DT(+, Pre), [β2m]$_T$ = 50 μg/mL (4.24 μM) and [ALB]$_T$ = 35 mg/mL (529 μM)) and dialysis patients after maintenance dialysis (DT(+, Post), [β2m]$_T$ = 20 μg/mL (1.69 μM) and [ALB]$_T$ = 38 mg/mL (574 μM)), respectively. The black arrows show the change in [β2m]$_T$ and [ALB]$_T$ by maintenance dialysis treatment. **i** Fluctuations of concentrations of the β2m monomer and serum albumin, temporary risk, *TR*, and accumulated risk, *AR*, as a function of the time. **j** Estimation of the onset of DRA by the accumulation of three risk factors at different conditions.

effects, accelerating amyloid fibril formation[34]; (ii) interactions with crowders, decelerating amyloid fibril formation[35]; and (iii) decrease in the diffusion constant, decelerating amyloid fibril formation[54]. The current results showed that serum albumin inhibited β2m amyloid formation by mechanism (ii): Weak nonspecific interactions between serum albumin and β2m monomer decreased the concentration of unfolded β2m through the "law of mass action"[23]. On the other hand, PEG molecules were inert to β2m monomer, slightly increasing the amount of amyloid fibrils by mechanism (i), but they retarded the kinetics by mechanism (iii). Serum albumin was primarily responsible for the inhibitory effects despite the weak nonspecific interactions. This could be achieved by a huge amount of serum albumin in a serum milieu: The concentration of serum albumin was ~45 mg/mL, being the highest among serum proteins with a total concentration of ~70 mg/mL.

If serum albumin had the ability to bind to serum proteins with a high affinity, most of them would fail to exert their biological functions because of strong trapping by serum albumin. Thus, it is inferred that the huge amount and weak affinity of serum albumin is crucial for

keeping the proteostasis network in a serum milieu. Previous study reported that serum albumin inhibited the amyloid fibril formation of amyloid-β (Aβ) peptide by trapping Aβ peptides in a nonfibrillar form[55]. Furthermore, other study reported that reduced serum albumin levels in vivo correlated with the onset of Alzheimer's disease[56], indicating that a decrease in the serum albumin concentration may be a risk factor for not only DRA, but also other amyloidoses.

Focusing on the inhibitory effects of serum albumin, we created a model of β2m amyloid formation in a serum milieu (Fig. 6a). We considered that the simplest stoichiometric binding existed between serum albumin (ALB) and β2m monomer in its native state (N) (Scheme 1), a two-state unfolding/folding between N and denatured state (D) (Scheme 2), and the seed-dependent amyloid elongation, where P represents polymeric amyloid fibrils (Scheme 3).

ALB : N ⇌ ALB + N, (Scheme 1)

N ⇌ D, and (Scheme 2)

D + P ⇌ P. (Scheme 3)

The equilibrium constants of Schemes 1–3 (i.e., dissociation of serum albumin-native β2m complex ($K_D^C$), folding/unfolding ($K_F$), and

amyloid elongation ($K_P$)) are represented by Eqs. (1)–(3):

$$K_D^C = \frac{[ALB][N]}{[ALB:N]}, \tag{1}$$

$$K_F = \frac{[N]}{[D]}, \text{ and} \tag{2}$$

$$K_P = \frac{[P]}{[P][D]_C} = \frac{1}{[D]_C}, \tag{3}$$

where [ALB:N], [ALB], [N], [D], and [P] refer to the molar concentrations of the serum albumin-native β2m complex, serum albumin, native β2m, denatured β2m, and amyloid fibrils, respectively, and $[D]_C$ refers to the solubility of denatured β2m monomers, which was 14.8 pM at 37 °C (Supplementary Note 8). Although the seed-dependent elongation assumes the presence of preformed fibrils, $K_P$ is independent of the concentration of seed fibrils. We considered that Scheme 3 and Eq. (3) are valid for amyloid fibril formation in general. $[D]_C$ is also referred to as the "critical concentration" because amyloid fibrils form when the concentration of denatured monomers exceeds $[D]_C$[26].

QCM measurements showed that $K_D^C$ does not depend notably on temperature (Fig. 5e and Supplementary Fig. 11). In contrast, $K_F$ and $K_P$ depend on temperature markedly as is evident from thermal unfolding of β2m monomers (Fig. 5h) and temperature-dependent amyloid formation[57], respectively. We assumed that temperature dependences of $K_F$ and $K_P$ as represented by the Gibbs free energy equations as reported previously[23,24,57] (Supplementary Notes 9 and 10, and Supplementary Fig. 13).

Before breakdown of supersaturation, amyloid fibril formation does not occur even when the concentration of denatured monomer exceeds $[D]_C$, where the Schemes 1 and 2 are valid. After breakdown of supersaturation, all three schemes are valid. For both before and after breakdown of supersaturation, we can calculate exactly the molar concentrations of all the species of Schemes 1–3 (Supplementary Note 10). In other words, supersaturation-dependent amyloid formation of β2m is represented as a function of β2m and serum albumin concentrations and temperature. These unified functions enabled the extrapolation of the results of HANABI assays performed at 60 °C to physiological conditions (Supplementary Fig. 14), as well as effects of change in β2m and serum albumin concentrations in patients.

Dependencies of respective species on total β2m and serum albumin concentrations in terms of concentrations ($[ALB:N]^S$ and $[D]^S$) or fractions ($F_{ALB:N}^S$ and $F_N^S$) at 37 °C are shown by red meshed surfaces in Fig. 6b, c and Fig. 6e, f, respectively, where superscript S indicates the value before breakdown of supersaturation. Figure 6c, f also shows $[D]^S$ (= $[N]^S/K_F$) and $F_D^S$. Moreover, the supersaturation ratio ($\sigma = [D]^S/[D]_C$), which is a direct indicator of the risk of amyloid fibril formation[58] and is proportional to $[D]^S$, is shown in Fig. 6c. The average values for non-dialysis controls and dialysis patients immediately before and after the maintenance dialysis treatment are indicated by blue, red, and green stars, respectively.

Upon increasing $[β2m]_T$ in the absence of $[ALB]_T$, the $\sigma$ value increased (Fig. 6c, f). However, this enhanced risk could be suppressed markedly by reducing D upon interaction with serum albumin. In other words, serum albumin works as a reservoir to reduce the risk of amyloid formation in terms of $\sigma$, or the law of mass action shifts the equilibrium to reduce D. Here, the $[D]_C$ value was estimated to be 14.8 pM (Supplementary Note 8), similar to the β2m concentration in healthy individuals and much lower than that in dialysis patients. However, no amyloid fibril was formed in healthy individuals because of the folded native structure and, moreover, the high free energy barrier of supersaturation even if they are denatured (Fig. 6d, g).

Although we considered the simplest case whereby interaction occurs with the native β2m monomers, it is possible that denatured β2m monomers and aggregates formed including oligomers and amyloid seeds are also adsorbed by serum albumin, further decreasing the risk. On the other hand, the interactions between serum albumin and β2m monomer in vivo are likely to be weaker than observed here in an isolated in vitro system because of varying competing components in a serum milieu.

After breakdown of supersaturation, dependencies of respective species on $[β2m]_T$ and $[ALB]_T$ at 37 °C in terms of concentrations ($[ALB:N]^E$, $[N]^E$ and $[P]^E$) or fractions ($F_{(ALB:N)}^E$ and $F_{(N)}^E$, and $F_{(P)}^E$), which are the relative concentration compared with $[β2m]_T$, are shown by blue meshed surfaces in Fig. 6b–d and Fig. 6e–g, respectively, where superscript E indicates the value after breakdown of supersaturation. Importantly, when $[D]^S > [D]_C$, $[D]^E = [D]_C$, independent of β2m and serum albumin total concentrations (Fig. 6c). Except for the serum albumin-bound and native β2m species determined by $[D]_C$, β2m monomers are converted to amyloid fibrils. The profiles suggest the marked impact of supersaturation on preventing amyloid fibril formation. The change in the concentrations of each β2m species is summarized in Supplementary Movie 1.

We further address the onset of DRA based on the supersaturation-limited amyloid fibril formation of β2m in the presence of serum albumin. Primary nucleation would be a rate-limiting step of amyloid fibril formation for hemodialysis patients[59]. Once nuclei are formed, amyloid fibrils propagate rapidly throughout the patient's body accelerated by secondary nucleation (i.e., fragmentations and lateral bindings), although it may take years until the onset of DRA. Thus, the potential for nucleation correlates with the risk of DRA onset. Based on the classical nucleation theory[58], the time for nucleation (i.e., lag time) shows an inverse correlation with the supersaturation ratio, $\sigma$, as ln(lag time) $\propto$ ln$^{-2}\sigma$ (details in Supplementary Note 11). Based on the supersaturation ratio, we investigated the relative change in the lag time as a function of $[β2m]_T$ and $[ALB]_T$ (Fig. 6h). As anticipated for dialysis patients (i.e., DT(+, Pre)), the higher β2m and lower serum albumin concentrations resulted in a shorter lag time.

During maintenance dialysis treatments, the relatively small β2m monomers (MW: ~11.8 kDa) pass through the pores of the dialysis membrane more than the larger serum albumin (MW: ~66.2 kDa). In parallel, maintenance dialysis removes a large amount of water from the patient's body, concentrating the serum components. Combined effects are a decrease and an increase in β2m and serum albumin concentrations, respectively (Fig. 6i). Consequently, the maintenance dialysis treatment reduces temporary risk (TR) for β2m amyloid fibril formation in a short term. However, the temporarily lowered risk gradually increases towards the next maintenance dialysis treatment: The β2m monomers newly produced accumulate and the concentration of serum albumin gradually decreases due to serum volume expansion by water uptake. It should be noted that the change in the concentrations of β2m monomer and serum albumin has no noticeable effects on the folding reaction of β2m monomer (Supplementary Note 9 and Supplementary Fig. 13b) because it is described as the fraction of each species. However, the supersaturation ratio, which is a relative concentration of the denatured monomer compared with the critical concentration, markedly fluctuates by the long-term and maintenance dialysis (Supplementary Note 9 and Supplementary Fig. 13c), causing a large change in the risk for amyloid fibril formation in vivo.

Considering the aforementioned fluctuation of TR for β2m amyloid fibril formation, we estimated the combined risk of DRA onset (Details in Supplementary Note 11). First, the degree of supersaturation at each time-point, $\sigma(t)$, was calculated as a function of β2m and serum albumin concentrations, which fluctuate during maintenance dialysis intervals (Fig. 6i). Second, from the $\sigma(t)$ value, the temporary risk, $TR(t)$,

of amyloid fibril formation was calculated as a function of time assuming that it is proportional to the inverse of the lag time, $TR(t) \propto \ln^2 \sigma(t)$ (Fig. 6i). Finally, the $TR(t)$ value was integrated over the whole dialysis vintage to calculate the accumulated risk, $AR(t)$, as $AR = \int_0^{DV} TR(t)\,dt$, where DV denotes the dialysis vintage in years (Fig. 6i).

Based on this model, we reproduced the onset of DRA (Fig. 6j) as reported in the cohort study conducted by Hoshino et al.[6]. For dialysis patients who developed DRA ($N = 2157$), the average dialysis vintage, β2m concentrations before and after maintenance dialysis treatments, and the serum albumin concentration before maintenance dialysis treatment were 18.2 years, $[\beta 2m]_T^{Pre} = 27.3\,\mu g/mL$ (2.31 μM), $[\beta 2m]_T^{Post} = 10.2\,\mu g/mL$ (0.86 μM), and $[ALB]_T^{Pre} = 36.6\,mg/mL$ (553 μM), respectively. Assuming that the maintenance dialysis increases the concentration of serum albumin to 3 mg/mL (i.e., 8%) by condensation (i.e., $[ALB]_T^{Post} = 39.6\,mg/mL$ (598 μM)), the $AR$ value would reach -3.9 at 18.2 years (black line in Fig. 6j).

According to this mechanism, the reason why the patients with increased β2m concentrations (primary risk) and a long dialysis vintage (secondary risk) do not necessarily develop DRA[4–8] is explained by the variation in $AR$ among individual patients. If the improvement of dialysis technology could reduce $[\beta 2m]_T^{Pre}$ and $[\beta 2m]_T^{Post}$ to 20 μg/mL (1.69 μM) and 5 μg/mL (0.42 μM), respectively, the onset of DRA would be delayed to 26.8 years (blue line in Fig. 6j), markedly contributing to improvement of the quality of life of dialysis patients. Meanwhile, if the serum albumin level was kept at a healthy level (e.g., $[ALB]_T^{Pre} = 42\,mg/mL$ (634 μM) and $[ALB]_T^{Post} = 45\,mg/mL$ (680 μM)), the onset of DRA would be delayed to 19.5 years (red line in Fig. 6j). The interaction of serum albumins with denatured β2m monomers and/or preformed aggregates, which we did not consider in this study, would further delay the onset of DRA. These considerations indicate that, although a decrease in the serum β2m concentration is the most effective way to prevent DRA, keeping the serum albumin concentration at a healthy level also contributes to the prevention of DRA by raising the inhibitory effects of the serum milieu on β2m amyloid fibril formation.

To understand the pathophysiology of amyloidosis, it is essential to study the effects of aberrant mutations[11,60] and post-translational modifications[61] at the single molecule level. Furthermore, atomic-level observation of amyloid fibrils by cryo-electron microscopy[16,62] is also important to understand the relationship between polymorphisms in amyloid fibrils and the phenotype of amyloidosis. On the other hand, elucidating the reactive milieu of amyloid fibril formation from a physicochemical point of view gives us a novel perspective on how to reduce the risk of developing amyloidosis. In general, maintaining a healthy proteostasis network around amyloidogenic proteins under supersaturation may contribute to the prevention of not only DRA but also other amyloidoses. Although we succeeded in identifying a new risk factor that regulates β2m monomer supersaturation in the pathogenesis of DRA, the direct mechanisms that disrupt the supersaturation barrier are still unknown and their clarification will be essential for the prevention of DRA and other amyloidoses.

In conclusion, using sera collected from dialysis patients, we found that serum components inhibited β2m amyloid fibril formation in a concentration-dependent manner. Although the inhibitory effects were deteriorated in long-term dialysis patients, the deteriorated inhibitory effects were ameliorated by the maintenance dialysis treatment in the short term. Among serum components, we found that serum albumin plays a major role in maintaining the serum proteostasis network and thus supersaturation of β2m monomer via weak nonspecific binding. Based on these results, we constructed the unified theoretical model to describe supersaturation-dependent amyloid formation of β2m as a function of β2m and serum albumin concentrations and temperature, allowing us to discuss amyloid fibril formation under physiological conditions by extrapolating the experimental results obtained under non-physiological conditions.

The theoretical calculation combined the elucidated risk factors (i.e., elevated β2m concentration, long dialysis vintage, and reduced serum albumin concentration) indicated the importance of monitoring temporary risk ($TR$) and accumulated risk ($AR$) by a methodology like HANABI-2000, enabling a supersaturation-aided strategy to prevent the development of amyloidosis in general.

## Methods

### Ethics statement
This study complies with all relevant ethical regulations. The study protocol adhered to the Declaration of Helsinki and was approved by the Central Ethics Committee of Niigata University (2018-0054/ 2022-0019) and Osaka University (T21069). All patients provided written informed consent. An opt-out option was also provided to allow patients to refuse study participation. We recruited donors of serum samples by informing the purpose of this study and collected serum samples from donors who agreed with the use of samples for this research.

### Recombinant β2m solutions
The recombinant β2m monomer with an additional methionine residue at the N terminus was expressed using *Escherichia coli* and purified as following procedures[63]. cDNA encoding β2m was amplified by PCR. The amplified DNA fragment was cloned into *E. coli* (BL-21 DE3, NIPPON GENE) expression vector pCold. Expression colonies were grown to an optical density of 0.5 at 600 nm in Luria-Bertani broth supplemented with 100 μg/mL ampicillin at 37 °C, and then, the protein synthesis was induced with 0.8 mM isopropyl-β-ᴅ-thiogalactoside at 16 °C overnight. The *E. coli* was harvested by a centrifugation procedure and lysed using BugBuster® (Merck) including 1 mg/mL lysozyme. After removal of supernatant by a centrifugation procedure, the pellet was dissolved in 20 mM Tris-HCl buffer (pH 8.0) supplemented with 8 M urea. To oxidize the β2m monomers, 10 mM 1,1′-Azobis(N,N-dimethylformamide) (TCI) dissolved in N,N-dimethylformamide was added to the solution. After dialysis overnight, the sample solution was applied onto DEAE Sepharose™ Fast Flow column (Cytiva) and was eluted with a linear gradient of 0–200 mM NaCl. The fractions containing β2m monomers confirmed by means of UV absorption and SDS-PAGE were collected and dialyzed overnight. The dialyzed sample solution was applied onto Resource™ Q column (Cytiva) and was eluted with a linear gradient of 0–100 mM NaCl. The fractions containing β2m confirmed by means of UV absorption and SDS-PAGE were again collected and dialyzed overnight. The purified β2m monomers were finally lyophilized. The molecular weight of β2m monomer was verified by means of SDS-PAGE and electrospray ionization mass spectrometry.

The lyophilized β2m monomer was stored at −20 °C directly prior to experiments. The monomer was dissolved into deionized water and filtrated using a pore filter with a pore diameter of 220 nm. The monomer solution was mixed with other chemicals to their final concentrations as follows: [β2m] = 1.0 mg/mL; [NaPi(pH 7.4)] = 20 mM; [NaCl] = 300 mM; and [ThT] = 5 μM. In the amyloid formation experiments, the serum samples were added to the recombinant β2m solution at a volume ratio between 0.3–15% (v/v).

### Collection and treatment of serum samples
**Comparison between dialysis patients and non-dialysis controls.** We recruited 30 patients undergoing dialysis treatment and 30 non-dialysis controls in a single center. In the dialysis patient group, dialysis treatments were 4–5 sessions conducted three times weekly with standard bicarbonate dialysate (Na⁺: 140 mEq/L, K⁺: 2.0 mEq/L, Ca²⁺: 2.75 mEq/L, Mg²⁺: 1.0 mEq/L, Cl⁻: 112.25 mEq/L, and HCO₃⁻: 27.5 mEq/L) and dialyzers with synthetic polysulfone membranes. Baseline data, including: age, sex, body mass index, cause of kidney disease, systolic blood pressure, and serum levels of urea nitrogen, creatinine, serum albumin, blood hemoglobin, and C-reactive protein, were measured in

both groups. Duration of dialysis treatment and single pool $Kt/V_{urea}$; were also reported in the dialysis patient group, and the estimated glomerular filtration rate (eGFR) was measured in the non-dialysis group. In the dialysis patient group, sera were collected before maintenance dialysis treatment. Continuous variables are expressed as medians (interquartile range).

Supplementary Data 1a shows the demographic and clinical characteristics of the non-dialysis controls ($N = 30$) and dialysis patients ($N = 30$), demonstrating typical data for end-stage kidney disease.

**Investigation of the effects of a single maintenance dialysis treatment on the degree of inhibitory effects.** We recruited 28 patients undergoing long-term dialysis treatment in multi-centers. Dialysis treatments were 4–5 h sessions conducted three times weekly with standard bicarbonate dialysate (Na$^+$: 140 mEq/L, K$^+$: 2.0 mEq/L, Ca$^{2+}$: 2.75 mEq/L, Mg$^{2+}$: 1.0 mEq/L, Cl$^-$: 112.25 mEq/L, and HCO$_3{}^{3-}$: 27.5 mEq/L) and dialyzers with synthetic polysulfone membranes. Baseline data, including: age, sex, body mass index, cause of kidney disease, systolic blood pressure, and serum levels of serum albumin, blood hemoglobin, calcium, phosphorus, parathyroid hormone, β2m, and C-reactive protein, were measured in both groups. Duration of dialysis treatment and single pool $Kt/V_{urea}$; were also reported. The patients underwent long-term dialysis treatment. Sera were collected before and after maintenance dialysis treatment. Continuous variables are expressed as medians (interquartile range).

Supplementary Data 1b shows the demographic and clinical characteristics of the long-term dialysis patients ($N = 28$), demonstrating usual data as end-stage kidney disease.

**Ultrasonic assays of amyloid fibril formation using HANABI-2000**
We used an originally developed ultrasonication system, HANABI-2000, which is optimized for accelerated amyloid fibril formation[36]. The 198-μL sample solutions were added to a 96-well plate (675096, Greiner) and sealed with plastic film (547-KTS-HC, Watson). The sample solutions inside the plate were irradiated with ultrasound at a frequency of ~30 kHz, an optimized frequency for accelerating amyloid formation[42]. During the experiments, ultrasonication was performed with duty cycles comprising 0.3-s irradiation and 30-s quiescence incubation. To monitor the kinetics of amyloid fibril formation, the ThT fluorescence intensity was measured with excitation and emission wavelengths of 450 and 490 nm, respectively. Fluorescence measurements were performed every 10 min until the end of the experiment, and the acquired data were recorded using SF6 software (Version 5.12.1, Corona Electric Co., ltd.). The temperature of the sample solution was kept at 60 °C. Scheme of the HANABI assay is described in Supplementary Fig. 2.

It should be noted that, in the HANABI assays using serum samples, the person who collected serum samples labelled the ID of samples and send the person who performed the experiments without disclosure of the patient information. After the series of the assay, the person who performed the experiment was informed which patient belongs to which group and started to analyze the experimental results. This blinding protocol was adopted to avoid the intervention of prejudices of the person in charge of the experiment.

**Circular dichroism (CD) spectrum measurements**
The CD spectra were measured using a CD spectrometer (JASCO Corp., J-820). The far- and near-UV CD spectra were acquired at wavelengths of 200–250 nm and 250–320 nm using quartz cells with light paths of 1 and 10 mm (JASCO Corp.), respectively. Thermal denaturation curves were obtained at wavelengths of 230 and 293.5 nm in the far- and near-UV regions, respectively, with a temperature change rate of 1 °C/min. The CD spectrum data were collected using Spectra Manager (Version 1.55.00, JASCO Corp.).

Regarding the amyloid fibrils, the sample solutions after HANABI assay had their β2m concentration adjusted to 0.15 mg/mL by dilution with deionized water. The 170-μL and 2-mL solutions were injected into a quartz cell with light paths of 1 mm and 10 mm, respectively.

**High performance reversed-phase chromatography**
First, the fibril solution was diluted 10-fold with dimethyl sulfoxide and incubated for 1 h at room temperature[64]. Then, 6 M HCl solution was added to completely dissolve the aggregates. For the analysis, we used a high-performance liquid chromatography system (GILSON) with a C4 300-Å column (5C4-AR-300, COSMOSIL). The analysis was performed three times for each sample. The chromatographic data were collected using Clarity (Version 7, DataApex).

**Transmission electron microscopy (TEM) observation**
A 10-μL aliquot of the sample solution was placed on a collodion-coated copper grid (Nisshin EM Co.) for 1 min, and the remaining solution was removed with filter paper. The sample was then stained with a 1% (w/v) uranyl acetate solution for 1 min. Finally, the surface of the grid was rinsed with deionized water. TEM observation was performed using an H-7650 transmission electron microscope (HITACHI) with an acceleration voltage of 80 kV. The micrographs were recorded using Hitachi H-7650 control software (Version 02.00 0103-03, Hitachi).

**Seeding experiments**
At first, the seed amyloid fibrils were prepared by ultrasonication at a β2m concentration of 1.0 mg/mL under the conditions without sera. Then, the seeds were added to the monomer solutions ([β2m] = 0.1 mg/mL, [NaPi(pH 7.4)] = 20 mM, [NaCl] = 300 mM, and [ThT] = 5 μM) at various seed concentrations. The amount of amyloid fibrils in aggregates prepared in the presence of sera was assayed by performing the same seeding reactions as above and comparing kinetics. The seeding experiments used intermittent shaking agitation with a cycle composed of 20-s shaking with revolution of 650 rpm and 580-s quiescent incubation. The temperature was 45 °C.

**Quartz crystal microbalance (QCM) measurements**
We investigated the interactions between β2m monomer and serum albumin using a wireless-electrodeless quartz crystal microbalance (QCM) biosensor[49,50]. AT-cut quartz resonators with a fundamental frequency of 65 MHz and an in-plane size of $1.8 \times 1.6$ mm$^2$ were used. Films of 2-nm chromium and 15-nm gold were deposited on both sides of the quartz plate. QCMs were first cleaned using piranha solution (98% H$_2$SO$_4$: 30% H$_2$O$_2$ = 7:3) and deionized water. Then, 10 mM self-assembled monolayer (SAM) molecules in absolute ethanol were injected, with subsequent incubation overnight at 4 °C to make the linker layer. After washing with absolute ethanol and deionized water, a mixture of 100 mM *N*-hydroxysuccinimide and 100 mM 1-(3-dimethylaminopropyl)-3-ethylcarbodiimide hydrochloride in deionized water was injected, followed by incubation at room temperature for 2 h to activate the SAM termini, and then QCMs were rinsed with deionized water. Subsequently, 150 μM β2m monomer in the buffer solution (20 mM NaPi, pH 7.4) was injected on both sides of QCM and incubated for 2 h at room temperature to immobilize the β2m monomer on the QCM surface. Finally, serum albumin or PEG solution with various concentrations was flowed using a micropump. The sensor cell was immersed in a water bath to keep the temperature constant during the measurement. The solution flow rate during measurement was ~0.5 mL/min. The QCM data were collected by a laboratory-developed software.

The obtained frequency change curves were fitted with a function of $\Delta f / f_0 = A(e^{-\alpha t} - 1)$, where $A$ and $\alpha$ are fitting constants. The obtained α values were plotted against the albumin concentration. Because the α value is written as $\alpha = k_a[\text{ALB}] + k_d$[51], the $k_a$ and $k_d$ values,

which are the rate constants for association and dissociation reactions, respectively, can be obtained from the slope and intercept of the linear regression line, respectively. The dissociation constant at each temperature was calculated by $K_D = k_d/k_a$. Here, the $K_D$ values were analyzed based on the assumption that the stoichiometry of binding between serum albumin and β2m monomer is 1:1.

## Solution nuclear magnetic resonance (NMR) measurements

$^{15}$N-uniformly labelled β2m solutions at 1.0 mg/mL containing 20 mM sodium phosphate pH 7, 100 mM NaCl, 2% $D_2O$, and varying concentrations of serum albumin were prepared for 2D $^1$H-$^{15}$N heteronuclear single quantum coherence (HSQC) experiments. The spectra were recorded on a Bruker Avance III 500 MHz spectrometer with a cryogenic probe at 37 °C. The spectrometer was operated at $^1$H frequency of 500.13 MHz and $^{15}$N frequency of 50.68 MHz. $^1$H-$^{15}$N HSQC spectra were acquired with 2048 complex points covering 7002.8 Hz for $^1$H and 192 complex points covering 1419.1 Hz for $^{15}$N. Resonance frequencies in the spectra were identified using the chemical shift lists on β2m[65]. NMR data were processed by TOPSPIN-NMR software (Version 4.1.1, Bruker Biospin) and analyzed using Sparky (Version 3.114)[66].

## Solubility measurements

To address β2m amyloid fibril formation based on the supersaturation-limited mechanism, we measured the solubility of β2m monomer using an ultracentrifugation method combined with an ELISA assay[37,57]. First, β2m solution containing [β2m] = 1.0 mg/mL, [NaPi (pH 7.4)] = 20 mM, [NaCl] = 300 mM, [ThT] = 5 μM was prepared, and then amyloid fibril formation was induced by ultrasonication using the HANABI-2000 instrument. After a sufficient reaction time over 20 h, the solution reached an equilibrium state with amyloid fibrils, which was confirmed by the saturation of ThT fluorescence intensity. The equilibrated solution was ultracentrifuged at $100,000 \times g$ for 1 h, and then, the concentration of β2m monomers in the supernatant was determined by the ELISA method (KGE019, Human beta 2-Microglobulin Parameter Assay Kit, R&D Systems). In the ELISA assay, optical density of the samples was recorded using a microplate reader (SH-9000, Corona Electric Co., ltd.) with SF6 software (Version 5.12.1, Corona Electric Co., ltd.).

## Statistics

All data are shown as plots with their mean ± standard deviation. Replicates are indicated in figure legends. The null hypothesis test was performed using a one-sided Student's $t$-test. The calculated $p$-value was shown as an exact value. To investigate correlation between two variants, the Pearson correlation coefficient was calculated.

## Reporting summary

Further information on research design is available in the Nature Research Reporting Summary linked to this article.

# Data availability

Source data are provided with this paper. All other data are available from the corresponding authors on request. Source data are provided with this paper.

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

## Acknowledgements

The authors would like to thank Ph. D. Masatomo So (IPR Osaka Univ.) for discussion. This study was performed as part of the Cooperative Research Program for the Institute for Protein Research, Osaka University (CR-21-02), and was supported by the Japan Society for the Promotion of Science (20K06580 to K.Y., 20K22628 to M.N., 21K19224 to Y.G., 22H02584 to Y.G., 22K14013 to K.N., and Core-to-Core Program A: Advanced Research Networks to Y.G.), Ministry of Education, Culture, Sports, Science and Technology (17H06352 to Y.G.), SENTAN from AMED (16809242 to Y.G.), and JKA and its promotion funds from AUTORACE to K.N.

## Author contributions

K.N., K.Y., S.Y., and Y.G. wrote the manuscript. K.N., K.Y., M.N., L.Z., and H.O. conducted the experiments. T.I., I.N., S.Y., and Y.G. managed the clinical samples and research ethics. K.N., K.Y., C.A., K.I., and H.M. analyzed the experimental data. F.G., H.N., S.Y., and Y.G. supervised the study.

## Competing interests

The authors declare no competing interests.

## Additional information

[1]Global Center for Medical Engineering and Informatics, Osaka University, Suita, Osaka 565-0871, Japan. [2]Graduate School of Human and Environmental Studies, Kyoto University, Yoshidahonmatsu-cho, Sakyo-ku, Kyoto 606-8316, Japan. [3]Department of Neurology, Graduate School of Medicine, Osaka University, Suita, Osaka 565-0871, Japan. [4]Graduate School of Engineering, Osaka University, Suita, Osaka 565-0871, Japan. [5]Division of Clinical Nephrology and Rheumatology, Graduate School of Medical and Dental Sciences, Niigata University, Niigata 951-8510, Japan. [6]Niigata University of Pharmacy and Applied Life Sciences, Niigata 956-8603, Japan. [7]Faculty of Medical Sciences, University of Fukui, Fukui 910-1193, Japan. [8]Present address: Graduate School of Engineering, Osaka University, Suita, Osaka 565-0871, Japan. [9]These authors contributed equally: Kichitaro Nakajima, Keiichi Yamaguchi, Masahiro Noji. ✉e-mail: yamamots@med.niigata.ac.jp; gtyj8126@protein.osaka-u.ac.jp

