## [Peer Review File · Nature Communications]

REVIEWER COMMENTS

Reviewer #1 (Remarks to the Author):

Nakajima and colleagues wrote a monumental work that convincingly presents the occurrence of an additional, new risk factor for the onset of DRA. Their conclusions can be considered paradigmatic for the onset of amyloidoses in general.

The work appears therefore relevant in the field of amyloidosis studies.

The workflow is very rigorously conducted through the possible checks that are necessary to support interpretation and conclusions. Also the stress on the supersaturation to explain the mechanisms triggering amyloidogenesis is well documented. This makes the interpretation soundly supported by literature evidence even though general consensus on the issue has not yet been reached.

The employed experimental methodologies are well established and the description of the procedures is well detailed.

Unfortunately the style the authors have chosen to present their results makes the reading very heavy. The impression is that the manuscript, as it is, would better suit another type of journal. For a general journal such as Nature Communications, one would expect a more descriptive text with a shift of all systematic arguments in the Supplementary Information, e.g. the schemes at the start of the Discussion.

So the advice is to simplify the Results description and the Discussion and to remove from the main text the systematic explanations. These can be reported in SI and replaced by short descriptive summaries. In alternative, the manuscript can be transferred as it is to a more specialized journal.

Typo: page 7, lines 3-4 from bottom

" ... of sera added, the lower the lag time (Figure 2D) and longer the ThT fluorescence"

should read

" ... of sera added, the longer the lag time (Figure 2D) and lower the ThT fluorescence"

Reviewer #2 (Remarks to the Author):

In the present manuscript, Nakajima and co-workers investigate how the formation of amyloid fibrils by b2-microglobulin (b2m), a phenomenon associated with dialysis-related amyloidosis (DRA), is influenced by human sera of both controls and patients and also before and after maintenance dialysis. The authors conclude that:

- i) Sera inhibited b2m amyloid formation.
- ii) inhibition was weaker for the patient's sera.
- iii) inhibition was more potent after dialysis, in the short term.
- iv) Serum albumin was a major contributor to the observed inhibitory effects.

Knowing how the aggregation of b2m occurs in its natural environment is still not clear and is an important issue to investigate, since, in the large majority of cases, DRA is reported for the wild type (WT) protein, whereas indeed this protein form is soluble in solution even at high concentrations. Unfortunately, although the experimental design of every single assay is accurate, in my view, the data together does not allow to advance significantly in explaining why and how b2m forms amyloids in vivo. This is related to my primary concerns in this work:

- b2m amyloid formation is induced in conditions that are far from physiological ones, requiring heating at 60 C and high ultrasonication. Although Th-T fluorescence did not occur without sonication, it is expected that in these conditions, b2m would be partially unfolded, or at least in a highly dynamic regime, which would recapitulate the conditions in human sera. This might imply that the conformation from which aggregation occurs in the body and the one from which it happens in the designed experiments might differ significantly and make uncertain the translational relevance of the work, despite its high biophysical standards.

- A major conclusion is that serum albumin is primarily responsible for the observed inhibitory effect of sera and that this effect responds to weak binding to b2m, which is compensated by the high concentrations of the protein in the serum. This protective activity of serum albumin against amyloid aggregation has been shown already proposed for the abeta amyloid peptide and, in this case, seems to be connected to both an interference with the nucleation stage and binding to protofibrils. Thus, it is interesting that this activity is also exerted for b2m. This said, a caveat of the present study is that binding assays are made under close to physiological conditions, i.e., 37 C, whereas aggregation is induced under much harsher conditions. Thus, is difficult to assess whether the observed binding affinity also applies under pro-aggregational conditions and, perhaps more importantly, which is the mechanism of action. Does serum albumin stabilize b2m, preventing further unfolding? Does it mask hydrophobic patches in a heat-induced dynamic conformation? Does it bind to the early or late b2m aggregates? In my opinion, these are important questions to address before a connection between the observed

albumin protective activity and binding to b2m can be asseverated to occur in the serum and could appear as a potential therapeutic or at least adjuvant alternative in DRA.

Reply to Reviewer #1

Reviewer's Comments #1-1: *Nakajima and colleagues wrote a monumental work that convincingly presents the occurrence of an additional, new risk factor for the onset of DRA. Their conclusions can be considered paradigmatic for the onset of amyloidoses in general. The work appears therefore relevant in the field of amyloidosis studies. The workflow is very rigorously conducted through the possible checks that are necessary to support interpretation and conclusions. Also the stress on the supersaturation to explain the mechanisms triggering amyloidogenesis is well documented. This makes the interpretation soundly supported by literature evidence even though general consensus on the issue has not yet been reached. The employed experimental methodologies are well established and the description of the procedures is well detailed.*

Author's Reply: The authors sincerely thank the reviewer for recognizing the importance of our work and for careful review. Based on your suggestions below, we revised our manuscript.

Revised sentences were highlighted in the attached PDF files, "Correction_Main.pdf" and "Correction_SI.pdf".

Reviewer's Comments #1-2: *Unfortunately the style the authors have chosen to present their results makes the reading very heavy. The impression is that the manuscript, as it is, would better suit another type of journal. For a general journal such as Nature Communications, one would expect a more descriptive text with a shift of all systematic arguments in the Supplementary Information, e.g. the schemes at the start of the Discussion. So the advice is to simplify the Results description and the Discussion and to remove from the main text the systematic explanations. These can be reported in SI and replaced by short descriptive summaries. In alternative, the manuscript can be transferred as it is to a more specialized journal.*

Author's Reply: Thank you very much for the important suggestion regarding the writing style. Following the reviewer's suggestion, we rewrote the manuscript to fascinate not only specialists in relevant fields but also general readers. We removed the detailed theoretical schemes from the main manuscript and moved to the supplementary information.

Reviewer's Comments #1-3: *Typo: page 7, lines 3-4 from bottom*

"... of sera added, the lower the lag time (Figure 2D) and longer the ThT fluorescence" should read

*"... of sera added, the longer the lag time (Figure 2D) and lower the ThT fluorescence" **Author's Reply:** We corrected these mistakes in the revised manuscript.*

Reply to Reviewer #2

Reviewer's Comments #2-1: *In the present manuscript, Nakajima and co-workers investigate how the formation of amyloid fibrils by β 2-microglobulin (β 2m), a phenomenon associated with dialysis-related amyloidosis (DRA), is influenced by human sera of both controls and patients and also before and after maintenance dialysis. The authors conclude that: i) Sera inhibited β 2m amyloid formation. ii) inhibition was weaker for the patient's sera. iii) inhibition was more potent after dialysis, in the short term. iv) Serum albumin was a major contributor to the observed inhibitory effects.*

Knowing how the aggregation of β 2m occurs in its natural environment is still not clear and is an important issue to investigate, since, in the large majority of cases, DRA is reported for the wild type (WT) protein, whereas indeed this protein form is soluble in solution even at high concentrations. Unfortunately, although the experimental design of every single assay is accurate, in my view, the data together does not allow to advance significantly in explaining why and how β 2m forms amyloids in vivo. This is related to my primary concerns in this work:

Author's Reply: The authors sincerely thank the reviewer for recognizing the importance of our work and for insightful comments. Based on your comments below, we revised our manuscript by conducting additional experiments, theoretical analyses, further discussion on suggested comments, and rewriting works. The revised points were highlighted in the attached PDF files, "Correction_Main.pdf" and "Correction_SI.pdf". Discussion on your important concerns, which are related to our experimental conditions and temperature dependence of the inhibition mechanism of serum albumin, is given below.

Considering the comment that "in the large majority of cases, DRA is reported for the wild type (WT) protein", we added a sentence in Introduction that "To date, only two cases of naturally occurring β 2m mutants have been reported: D76N in non-dialysis patients (Valleix et al. N. Engl. J. Med. Vol. 366, 2276 (2012)) and V27M in a DRA patient (Mizuno et al. Amyloid Vol. 28, 42 (2021))".

Reviewer's Comments #2-2: *β 2m amyloid formation is induced in conditions that are far from physiological ones, requiring heating at 60 C and high ultrasonication. Although Th-T fluorescence did not occur without sonication, it is expected that in these conditions, β 2m would be partially unfolded, or at least in a highly dynamic regime, which would recapitulate the conditions in human sera. This might imply that the conformation from which aggregation occurs in the body and the one from which it happens in the designed experiments might differ significantly and make uncertain the translational relevance of the work, despite its high biophysical standards.*

Author's Reply: Thank you very much for these important comments regarding the experimental design. As you pointed out, in this study, we performed the accelerative amyloid fibril formation assay under non-physiological conditions, such as at high temperature (60 °C) and under ultrasonication. Previously, experiments conducted under non-physiological high temperatures have been adopted to investigate folding (i.e., formation of native proteins) and misfolding (i.e., formation of amyloid fibrils) mechanisms and have contributed to the elucidation of those reaction mechanisms. Since thermodynamic parameters obtained under extreme conditions can be extrapolated back to the reaction mechanism under physiological conditions, they have been essential to discussing protein folding and misfolding reactions. Furthermore, we have demonstrated in previous studies that mechanical agitations including ultrasonication and stirring are an effective way to induce amyloid fibril formation by reducing the energy barrier toward the primary nucleation, dramatically decreasing the time for amyloid formation assays (M. Noji et al., *Commun. Biol.* Vol. 4, 120 (2021), and K. Nakajima et al., *ACS Chem. Neurosci.* Vol. 12, pp. 3456-3466 (2021)). Based on these backgrounds, we investigated the effects of human sera on the α 2m amyloid fibril formation under nonphysiological conditions and discussed potential risk factors for the onset of DRA. We are confident of the importance of this biophysical approach to revealing the mechanism of amyloid fibril formation under physiological conditions.

However, our descriptions and experimental data in the previous manuscript were insufficient to link amyloid fibril formation under experimental and physiological conditions. To make up for the shortage of these points, we experimentally obtained thermodynamic parameters regarding temperature dependence of the stability of α 2m monomers using nearUV circular dichroism (CD) spectroscopy (Fig. S12 and Table S4), allowing us to link amyloid fibril formation under our experimental and physiological conditions. The results showed that ~14.3% α 2m monomers were unfolded at 60 °C. Furthermore, the far-UV CD spectroscopy (Fig. 1b,c) and quartz-crystal microbalance (QCM) analysis were performed at high temperatures in addition to 37 °C (Fig. S11). The results indicated that the interaction between α 2m monomer and serum albumin was unchanged even at high temperatures. Finally, we added temperature as a variable to the theoretical model of serum albumin-independent amyloid fibril formation (Figs. S13 and S14). The unified model allowed us to express amyloid fibril formation in the presence of serum albumin at different temperatures by a combination of simple physicochemical equations. The temperature dependence showed that, although the stability of the native α 2m monomer does not increase markedly upon interaction with serum albumin, the slight decrease in the denatured α 2m monomer decreases the supersaturation ratio markedly, thus reducing the risk of amyloid fibril formation. The revised manuscript emphasizes the importance of discussion on

amyloid formation based on the supersaturation-limited mechanism. These revisions clearly deepened the discussion in our paper and led to a more comprehensive understanding of the effects of serum albumin. See also our response to Comments #2-3-3.

Reviewer's Comments #2-3: *A major conclusion is that serum albumin is primarily responsible for the observed inhibitory effect of sera and that this effect responds to weak binding to $\square 2m$, which is compensated by the high concentrations of the protein in the serum.*

Author's Reply: As the reviewer kindly summarized, we insist the importance of serum albumin as an environmental factor determining the onset of DRA, which is highly influenced by dialysis treatment. Concerns regarding this point are separately answered one by one by dividing the reviewer's comments into six small questions.

Reviewer's Comments #2-3-1: *This protective activity of serum albumin against amyloid aggregation has been shown already proposed for the abeta amyloid peptide and, in this case, seems to be connected to both an interference with the nucleation stage and binding to protofibrils. Thus, it is interesting that this activity is also exerted for $\square 2m$.*

Author's Reply: Regarding this comment, we checked several previous reports. Stanyon and Viles performed an in-vitro experimental study and reported that serum albumin inhibits the amyloid beta (A \square) fibril formation by which serum albumin traps A \square peptide in a nonfibrillar form through weak binding (H. F. Stanyon and J. H. Viles, J. Biol. Chem. Vol. 287, pp. 2816328168 (2012)). Furthermore, Yamamoto and his coworkers performed a clinical study and reported that a decrease in the serum concentration of the serum albumin-A \square complex correlates with the prevalence of Alzheimer's disease (K. Yamamoto *et al.*, Geriatr. Gerontol. Int. Vol. 14, pp. 716-723 (2014)). These reports insist that reduced serum albumin levels in vivo correlate with the onset of Alzheimer's disease, consistent with our results, implying that reduced serum albumin levels would be a common risk factor for the onset of a variety of amyloidoses. We added these descriptions in the revised manuscript to address the commonality of serum albumin effects on the onset of amyloidoses.

Reviewer's Comments #2-3-2: *This said, a caveat of the present study is that binding assays are made under close to physiological conditions, i.e., 37 C, whereas aggregation is induced under much harsher conditions. Thus, is difficult to assess whether the observed binding affinity also applies under pro-aggregational conditions and, perhaps more importantly, which is the mechanism of action.*

Author's Reply: To confirm whether the measured binding affinity at 37 °C is applicable to the amyloid fibril formation at 60 °C or not, we additionally evaluated the binding affinity at

high temperatures (i.e., 37, 50, and 60 °C) by means of QCM measurement (Fig. S11). The results showed that the dissociation constant between $\alpha 2m$ monomer and serum albumin remained on the order of ~ 100 μM independent of the temperature, demonstrating that the inhibitory effects of serum albumin on $\alpha 2m$ amyloid fibril formation is caused by weak nonspecific binding, and the similar effects will be common to amyloid formation under physiological conditions. We added the results of the additional QCM experiments and relevant discussion in the revised manuscript.

Reviewer's Comments #2-3-3: *Does serum albumin stabilize $\alpha 2m$, preventing further unfolding?*

Author's Reply: To discuss this point, we additionally performed near-UV CD measurements to evaluate the melting point (T_m) of the native $\alpha 2m$ monomer (Fig. 5h). We added serum albumin to the $\alpha 2m$ monomer solution with the stoichiometry $\alpha 2m$:ALB between 10:1-1:1. Within these stoichiometries, the addition of serum albumin increased the T_m value of the native $\alpha 2m$ monomer ~ 1 °C, showing a slight stabilization. Furthermore, we performed a theoretical approach to address the effects of serum albumin on the stability of $\alpha 2m$. From the melting curve of $\alpha 2m$ monomer without serum albumin, thermodynamic parameters were obtained. Theoretical calculation with these parameters and thermodynamic equations revealed that serum albumin stabilized native $\alpha 2m$ monomers, but the degree of stabilization was subtle and was approximately 1 °C in the T_m value, consistent with the experimental result. This fact indicates that the binding of serum albumin to native $\alpha 2m$ monomers has a slight effect on the folding reaction because the folding reaction is determined by the fraction of denatured monomers. However, the binding highly inhibits amyloid fibril formation through the decrease in the supersaturation ratio of denatured monomers, because amyloid fibril formation is dominated by the concentration of denatured monomers relative to its solubility. We added these descriptions and results of additional experiments in the revised manuscript. See also our response to Comments #2-2.

Reviewer's Comments #2-3-4: *Does it mask hydrophobic patches in a heat-induced dynamic conformation?*

Author's Reply: Due to experimental limitations, we could not perform the NMR measurement at high temperatures. However, we infer that the interactions between $\alpha 2m$ monomer and serum albumin are nonspecific without specific binding sites even at high temperatures because there was no difference in the dissociation constant at 37 and 60 °C, as shown by QCM measurements. We added these descriptions in the revised manuscript.

Reviewer's Comments #2-3-5: *Does it bind to the early or late β 2m aggregates?*

Author's Reply: Serum albumin possibly binds not only to β 2m monomers but also to β 2m oligomeric amorphous aggregates and/or amyloid fibrils through the nonspecific interaction. We here ignored the binding between serum albumin and formed aggregates, since the primary nucleation of amyloid fibrils is the rate-limiting step during amyloid fibril formation. However, these bindings possibly further decrease the risk for amyloid fibril formation. We added these descriptions in the revised manuscript.

Reviewer's Comments #2-3-6: *In my opinion, these are important questions to address before a connection between the observed albumin protective activity and binding to β 2m can be asseverated to occur in the serum and could appear as a potential therapeutic or at least adjuvant alternative in DRA.*

Author's Reply: We would like to express our deepest gratitude to the reviewer for their valuable comments. As a summary of the revisions, we performed additional experiments to demonstrate that the inhibitory effects of serum albumin on β 2m amyloid fibril formation observed in the HANABI assays is maintained under physiological conditions. Furthermore, we constructed the improved theoretical model that is unified the temperature effect, allowing us to express amyloid fibril formation by a combination of simple physicochemical equations as a function of temperature and concentrations of β 2m monomer and serum albumin. Using this model, we linked amyloid formation between experimental and physiological conditions. We believe that these revisions provide evidence that, under physiological conditions, serum albumin plays an important role in serum proteostasis network in protecting the dialysis patients from amyloid fibril formation of β 2m, that is, the onset of dialysis-related amyloidosis.

REVIEWERS' COMMENTS

Reviewer #1 (Remarks to the Author):

Please refer to the attached files

After reading the revised version of the manuscript by Nakajima and colleagues, I can only confirm my previous positive judgement on the work and my favorable opinion for the publication.

I appreciate the efforts of the authors to simplify the reading of their work by shifting an entire section of the Discussion to Supplementary Information, although the additions that were introduced in response to the objections raised by the other reviewer elided the effect, so that the manuscript is now even longer than the original one and still demands careful reading. However, I must admit that it is not easy to condense too much the subject without turning into simplistic and possibly superficial version. Therefore I would accept the manuscript for publication as it is.

There are a few typos or language corrections that should be considered for the publication (no further revision is required from this reviewer, however).

- Page 4, lines 13-14: probably something is missing in the sentence "We have studied β 2m amyloid fibril formation from the physicochemical viewpoint that amyloid fibril formation is similar". Do the author mean "We have studied beta2m amyloid fibril formation from the physicochemical viewpoint concluding that amyloid fibril formation is similar"?

- Page 4, penultimate line: the expression "Under supersaturated condition above solubility, an unknown trigger" appears redundant. I would drop "above solubility". Otherwise it seems possible to reach a supersaturated condition below the solubility limit.

- Page 17, line 19: "when $[D]S > [D]C$, $[D]E = [D]C$ and $\sigma = 1$, independent". This seems in formal contrast with the definition of sigma (given on page 16), unless one states that, under supersaturation break conditions, sigma corresponds to $[D]E/[D]C$. The issue can be addressed explicitly in Supplementary Information.

- Page 21, line 15: "... by extrapolating the experimental results under non-physiological conditions". I would add "obtained", i.e. "... by extrapolating the experimental results obtained under non-physiological conditions" to make clear that the extrapolation started from results collected under non-physiological conditions to discuss, by extrapolation, the amyloid formation under physiological conditions.

Reviewer #2 (Remarks to the Author):

The authors have done a significant and accurate effort to reply/clarify my concerns. The novel experiments and discussions dissipated my doubts, as long as they were experimentally addressable, and added biological relevance to the work.

Reviewer #3 (Remarks to the Author):

Nakajima et al presented results about the influence of serum albumin on the onset of dialysis-related amyloidosis (DRA). The serum albumin concentration in serum was identified as the tertiary risk factor. Many experiments using different methods and advanced techniques have been carried out to confirm this observation. Additionally, the theoretical model was developed for the evaluation of β 2m amyloid fibril formation dependence on β 2m and serum albumin concentrations and on the temperature. Experimental results were obtained from testing over 100 patients. The obtained results are scientifically interesting and have a high impact on the understanding of the impact of risk factors for dialysis-related amyloidosis.

A very important question related to experimental conditions was raised because at 60 °C β 2m

monomers are at least partly unfolded. Therefore, classical methods used for the calculation of thermodynamics characteristics of interactions are not completely correct. But authors did their best to explain the applicability of applied thermodynamics/kinetics-based methods and improved the description and assessment of the presented experimental data. Therefore, I think at current-state-of-the-art of physicochemical evaluation of protein-protein interaction these methods still can be applied. The conclusions are supported by the experimental results.

Minor comments

Why for the QCM measurements PEG was selected? QCM-D measurements could provide more information about the formed layers' properties at different experimental conditions.

Figure 5D should be corrected. It is difficult to understand which line belongs to which experimental conditions. Please change the colors and style of lines.

Reply to Reviewer #1

Reviewer's Comments #1-1: *After reading the revised version of the manuscript by Nakajima and colleagues, I can only confirm my previous positive judgement on the work and my favorable opinion for the publication. I appreciate the efforts of the authors to simplify the reading of their work by shifting an entire section of the Discussion to Supplementary Information, although the additions that were introduced in response to the objections raised by the other reviewer elided the effect, so that the manuscript is now even longer than the original one and still demands careful reading. However, I must admit that it is not easy to condense too much the subject without turning into simplistic and possibly superficial version. Therefore I would accept the manuscript for publication as it is. There are a few typos or language corrections that should be considered for the publication (no further revision is required from this reviewer, however).*

Author's Reply: The authors sincerely appreciate the reviewer for the careful review and positive comments on our work. As you kindly mentioned, although we tried to condense important matters by descriptive sentences, it was too difficult to shorten the whole manuscript because of the addition of several important experimental results to the revised version. In this final revision, we revised several sentences including the ones you mentioned. The revised sentences were highlighted in the PDF file 'correction.pdf'.

Reviewer's Comments #1-2: *Page 4, lines 13-14: probably something is missing in the sentence "We have studied α 2m amyloid fibril formation from the physicochemical viewpoint that amyloid fibril formation is similar". Do the author mean "We have studied beta2m amyloid fibril formation from the physicochemical viewpoint concluding that amyloid fibril formation is similar"?*

Author's Reply: We revised this sentence following your point.

Reviewer's Comments #1-3: *Page 4, penultimate line: the expression "Under supersaturated condition above solubility, an unknown trigger" appears redundant. I would drop "above solubility". Otherwise it seems possible to reach a supersaturated condition below the solubility limit.*

Author's Reply: We deleted the terms "above solubility" in the revised version.

Reviewer's Comments #1-4: *Page 17, line 19: "when $[D]S > [D]G$, $[D]E = [D]G$ and $\sigma = 1$, independent". This seems in formal contrast with the definition of sigma (given on page 16), unless one states that, under supersaturation break conditions, sigma corresponds to $[D]E/[D]C$. The issue can be addressed explicitly in Supplementary Information.*

Author's Reply: Thank you very much for this important comment. In this sentence, we intended to explain that the concentration of the denatured monomer reaches its solubility in equilibrium after the breakdown of supersaturation, independent of total concentrations of

2m monomers and serum albumin. The supersaturation ratio Ω is a variable describing the solution condition before the breakdown of supersaturation. Thus, the description you pointed out was inaccurate, and we deleted the description using the Ω value in this sentence in the revised version.

Reviewer's Comments #1-5: *Page 21, line 15: "... by extrapolating the experimental results under non-physiological conditions". I would add "obtained", i.e. "... by extrapolating the experimental results obtained under non-physiological conditions" to make clear that the extrapolation started from results collected under non-physiological conditions to discuss, by extrapolation, the amyloid formation under physiological conditions.*

Author's Reply: We agree this comment. We added the term 'obtained' between 'results' and 'under' in the revised version.

Reply to Reviewer #2

Reviewer's Comments #2-1: *The authors have done a significant and accurate effort to reply/clarify my concerns. The novel experiments and discussions dissipated my doubts, as long as they were experimentally addressable, and added biological relevance to the work.*

Author's Reply: The authors sincerely appreciate the reviewer for many valuable comments regarding the biological relevance of our experimental results and the validity of the experimental conditions. Owing to these comments, we could consider the additional experiments to improve discussion on the experimental results. In this final revision, we corrected several sentences following the comments given by other reviewers and editorial staffs. The revised sentences were highlighted in the PDF file 'correction.pdf'.

Reply to Reviewer #3

Reviewer's Comments #3-1: *Nakajima et al presented results about the influence of serum albumin on the onset of dialysis-related amyloidosis (DRA). The serum albumin concentration in serum was identified as the tertiary risk factor. Many experiments using different methods and advanced techniques have been carried out to confirm this observation. Additionally, the theoretical model was developed for the evaluation of β 2m amyloid fibril formation dependence on β 2m and serum albumin concentrations and on the temperature. Experimental results were obtained from testing over 100 patients. The obtained results are scientifically interesting and have a high impact on the understanding of the impact of risk factors for dialysis-related amyloidosis.*

A very important question related to experimental conditions was raised because at 60 °C β 2m monomers are at least partly unfolded. Therefore, classical methods used for the calculation of thermodynamics characteristics of interactions are not completely correct. But authors did their best to explain the applicability of applied thermodynamics/kinetics-based methods and improved the description and assessment of the presented experimental data. Therefore, I think at current-state-of-the-art of physicochemical evaluation of protein-protein interaction these methods still can be applied. The conclusions are supported by the experimental results.

Author's Reply: The authors sincerely appreciate the reviewer for reviewing our manuscript carefully and recognizing the importance of our research. We revised our manuscript following your comments regarding the section on QCM measurement. The replies to the reviewer's comments were given below point-by-point. The revised sentences were highlighted in the PDF file 'correction.pdf'.

Reviewer's Comments #3-2: *Why for the QCM measurements PEG was selected?*

Author's Reply: PEG with termini of a hydroxyl group is often adopted as an inert macromolecular crowder for amyloid fibril formation experiments or a blocking reagent in QCM measurement because of notably low affinity for proteins. Then, PEG molecular was selected as a negative control for serum albumin, allowing us to demonstrate that the interaction between serum albumin and β 2m monomers is a weak interaction but certainly exists, and this interaction plays an important role in the inhibition of amyloid fibril formation. These explanations were added in the revised manuscript.

Reviewer's Comments #3-3: *QCM-D measurements could provide more information about the formed layers' properties at different experimental conditions.*

Author's Reply: As the reviewer pointed out, QCM-D measurements could provide important insights into the viscoelastic properties of biomolecular membranes formed on the QCM surface by measuring dissipation of resonant oscillation of a quartz crystal sensor.

These investigations are highly important to reveal how membranes composed of amyloid fibrils mechanically damage biological tissue in vivo. In fact, our research group has investigated the change in viscoelastic properties of the amyloid-fibril membrane during the formation process (Y.-T. Lai et al., Langmuir 2018, 34, 5474-5479.). However, in the present study, we focused on evaluating the dissociation constants of interaction between serum albumin and β 2m monomers at various temperatures. We then judged that the QCM measurement is more suitable for this purpose rather than QCM-D measurement.

Reviewer's Comments #3-4: *Figure 5D should be corrected. It is difficult to understand which line belongs to which experimental conditions. Please change the colors and style of lines.*

Author's Reply: We revised this figure in the revised manuscript.